# Organizational Principles of the Centrifugal Projections to the Olfactory Bulb

**DOI:** 10.3390/ijms24054579

**Published:** 2023-02-26

**Authors:** Li Wang, Xiangning Li, Fengming Chen, Qing Liu, Fuqiang Xu

**Affiliations:** 1State Key Laboratory of Magnetic Resonance and Atomic and Molecular Physics, Key Laboratory of Magnetic Resonance in Biological Systems, Wuhan Center for Magnetic Resonance, Innovation Academy for Precision Measurement Science and Technology, Chinese Academy of Sciences, Wuhan 430071, China; 2Wuhan National Laboratory for Optoelectronics, Huazhong University of Science and Technology, Wuhan 430074, China; 3Center for Excellence in Brain Science and Intelligence Technology, Chinese Academy of Sciences, Shanghai 200031, China; 4College of Life Sciences, Wuhan University, Wuhan 430072, China; 5Shenzhen Key Laboratory of Viral Vectors for Biomedicine, Shenzhen Key Laboratory of Quality Control Technology for Virus-Based Therapeutics, Guangdong Provincial Medical Products Administration, NMPA Key Laboratory for Research and Evaluation of Viral Vector Technology in Cell and Gene Therapy Medicinal Products, The Brain Cognition and Brain Disease Institute (BCBDI), Shenzhen Institute of Advanced Technology, Chinese Academy of Sciences, Shenzhen-Hong Kong Institute of Brain Science-Shenzhen Fundamental Research Institutions, Shenzhen 518055, China; 6University of Chinese Academy of Sciences, Beijing 100049, China

**Keywords:** centrifugal projections, olfactory bulb, mitral/tufted cells, granule cells, virus-mediated tracing

## Abstract

Centrifugal projections in the olfactory system are critical to both olfactory processing and behavior. The olfactory bulb (OB), the first relay station in odor processing, receives a substantial number of centrifugal inputs from the central brain regions. However, the anatomical organization of these centrifugal connections has not been fully elucidated, especially for the excitatory projection neurons of the OB, the mitral/tufted cells (M/TCs). Using rabies virus-mediated retrograde monosynaptic tracing in Thy1-Cre mice, we identified that the three most prominent inputs of the M/TCs came from the anterior olfactory nucleus (AON), the piriform cortex (PC), and the basal forebrain (BF), similar to the granule cells (GCs), the most abundant population of inhibitory interneurons in the OB. However, M/TCs received proportionally less input from the primary olfactory cortical areas, including the AON and PC, but more input from the BF and contralateral brain regions than GCs. Unlike organizationally distinct inputs from the primary olfactory cortical areas to these two types of OB neurons, inputs from the BF were organized similarly. Furthermore, individual BF cholinergic neurons innervated multiple layers of the OB, forming synapses on both M/TCs and GCs. Taken together, our results indicate that the centrifugal projections to different types of OB neurons may provide complementary and coordinated strategies in olfactory processing and behavior.

## 1. Introduction

We live in an ever-changing environment, which makes changes in perception dependent on expectations, experiences, emotions, attention, etc. The centrifugal projections from the central brain areas to the early sensory areas are associated with information coding and modulation, including relaying information pertaining to the internal states of animals and sharpening of sensory representations [1,2,3,4,5]. The olfactory bulb (OB), the first relay center of the olfactory system, is a perfect model for studying the underlying mechanisms of centrifugal modulation, due to its relatively simple and unique anatomical structure. The OB receives primary afferent input from the sensory neurons, then processes and sends information to the primary olfactory cortex. Meanwhile, the OB receives a surprisingly large number of centrifugal inputs from the central brain regions [6,7,8], modulating the olfactory processing and behavior [1,9,10]. Dissecting the organizational principles of centrifugal projections to the OB is essential for shedding light on the coding and modulating strategies of the olfactory system.

Centrifugal fibers arising from the central brain areas to the OB are distributed in a layer-specific pattern [11,12,13]. Most of them terminate in the granule cell layer (GCL) and make synaptic contacts with the inhibitory interneurons, mainly the granular cells (GCs) [13,14,15], which, in turn, indirectly regulate the activity of excitatory projection neurons, namely the mitral/tufted cells (M/TCs) [16,17,18]. In addition, some centrifugal fibers also terminate in more superficial layers of the OB such as the mitral cell layer (MCL) and the external plexiform layer (EPL), where they perhaps directly form synapses with the M/TCs [19,20,21,22]. The centrifugal projections to the OB with different cellular sources and/or targets differ in their functional effects [1,10,21,23], partially caused by anatomical differences. However, previously, more attention has been focused on the centrifugal input to the GCs [8,24,25,26,27]; the organizational principles of the centrifugal input to the M/TCs remain largely unknown.

By combining virus-mediated tracing, whole-brain imaging, and immunohistochemistry, we elucidated the organizational features of the centrifugal input to the M/TCs. Although they are similar to GCs, which are predominantly innervated by the anterior olfactory nucleus (AON), piriform cortex (PC), and basal forebrain (BF), the M/TCs received proportionally less input from the AON and PC, and more input from the BF. Moreover, we found that individual BF cholinergic neurons can co-innervate both M/TCs and GCs. The diversified input types, patterns, and strengths reflect different coding and modulating strategies and therefore different functional roles of the centrifugal projections to different types of OB neurons.

## 2. Results

### 2.1. Brain-Wide Distribution of the Centrifugal Input to the OB

To trace the direct input of the M/TCs in the OB, RV-based retrograde monosynaptic viruses were injected into the MCL/EPL of the medial or dorsolateral OB in Thy1-Cre mice [28] (Figure 1A and Appendix A). Starter neurons predominantly resided in the MCL of the injected OB, with a small subset in the EPL (Figure 1B,C and Appendix A). As well as the adjacent AON and PC, the BF, medial amygdalar nucleus (MEA), and locus coeruleus (LC) were also efficiently labeled (Figure 1D and Appendix A).

The GCL is one of the major recipients of centrifugal projections in the OB [12,15,29] and contains the most abundant population of inhibitory interneurons, the GCs. In order to compare the input patterns between the excitatory projection neurons (M/TCs) and the inhibitory interneurons (GCs), a G-deleted rabies virus, RV-ΔG-mCherry, was injected into the GCL of the medial OB in C57BL/6J mice (Appendix A). We found that, similar to those of the M/TCs, the input neurons of the GCs were numerous in the AON, PC, and BF, but absent in the olfactory tubercle (OT) (Appendix A), consistent with previous tracing studies [8,24,25]. In addition, we also found input neurons in some secondary olfactory areas, including the lateral entorhinal cortex (LEC) and the ventral CA1 part of the hippocampal region (HIP) (Appendix A), consistent with a previous report [24].

We counted all input neurons from the anterior forebrain to the posterior medulla except the injected OB, and quantified the weighted input. M/TCs in the medial and dorsolateral OB received generally similar input from the central brain regions (Figure 2A–C). However, whether the dorsal and ventral M/TCs receive centrifugal input in similar ways needs to be further investigated. Next, we principally focused on an analysis of the medial M/TC and medial GC tracing groups. We found that the input to both M/TCs and GCs were ipsilaterally dominant, with the ratio of the contralateral input to the ipsilateral input being around 10%, and this was slightly higher in the M/TC tracing group (*p* = 0.017; Figure 2B). For both the M/TCs and GCs, the most prominent input came from the ipsilateral AON, accounting for about a half of the total, followed by the ipsilateral BF, the ipsilateral PC, and the contralateral AON (Figure 2C). Our data also revealed some specificity, namely that M/TCs received lower proportional input from the ipsilateral PC (*p* ˂ 0.001), but higher input from the ipsilateral BF and the contralateral AON (ipsilateral BF, *p* = 0.001; contralateral AON, *p* = 0.013; Figure 2C) than GCs.

### 2.2. Centrifugal Input from the Primary Olfactory Areas to the OB

The AON, the largest source of centrifugal input to the OB [25], can be subdivided into anterior olfactory nucleus pars externa (AONpE) and pars principalis (AONpP) [29]. Both the M/TCs and GCs were innervated by the bilateral AONs (Figure 3A,B). Input neurons targeting the M/TCs concentrated posteriorly, whereas those targeting the GCs were scattered broadly (Figure 3C,D). Moreover, both the M/TCs and GCs received predominant input from the AONpP (*p* < 0.001), and the GCs were also innervated by the contralateral AONpE (Figure 3E,F), consistent with previous output tracing data from the AON [19,20]. Immunohistochemical staining revealed that input neurons in the bilateral AONs were mostly CaMKII-immunoreactive for both groups (Figure 3G–I).

The PC, another key input source of the OB, is the largest olfactory cortex and has a trilaminar structure that can be divided into the anterior and posterior piriform cortex (APC and PPC) [30,31]. Similar to the GCs, the PC input to the M/TCs was ipsilateral (Figure 4A,B). Although both M/TCs and GCs were preferentially innervated by the APC (Figure 4D), the PC neurons targeting the M/TCs were concentrated anteriorly and preferentially in Layer III, whereas the ones targeting GCs were scattered broadly and mainly in Layer II (Figure 4C,E). Immunohistochemical staining revealed that, for both groups, the input neurons in the PC were mostly CaMKII-immunoreactive (Figure 4F–H).

### 2.3. Centrifugal Input from the Neuromodulatory Areas to the OB

The BF, the major input source from the neuromodulatory areas to the OB, is considered to be central to cognitive functions [23,32]. Both the M/TCs and GCs are heavily innervated by the ipsilateral BF, with the majority of input neurons located in the diagonal band nucleus (NDB) and the magnocellular preoptic nucleus (MA) (Figure 5A,B). Moreover, the BF input to the M/TCs and GCs occupied similar spatial locations across all the AP, medial–lateral (ML), and dorsal–ventral (DV) axes (Figure 5C), and shared a comparable cellular composition except that the proportion of ChAT^+^/GAD^−^ input neurons was higher in the M/TC tracing group (*p* = 0.029; Figure 5D–F). Therefore, there is the possibility that M/TCs and GCs are innervated by the BF cholinergic neurons in same population. To prove this, a dual-retrograde tracing experiment was performed to label the input of the M/TCs and GCs simultaneously (Figure 5G). We identified some overlapping of co-labeled neurons in the BF (Figure 5H), and about one-third of them were co-expressed with ChAT (Figure 5I).

To further identify whether one single cholinergic neuron co-innervated the M/TCs and GCs, we injected Cre-dependent rAAV2/9-CAG-FLEX-GFP into the BF of Chat-ires-Cre mice (Figure 6A) and performed whole-brain imaging of the fluorescently labeled cholinergic neurons. Among them, six OB-projecting cholinergic neurons were selected and reconstructed (Figure 6B). These neurons spread widely to the primary olfactory areas (Figure 6B,C). Although the projection patterns in the OB were distinct (Figure 6B,E), these individual cholinergic neurons sent their axon terminals to multiple layers of the OB, including the MCL, EPL, and GCL (Figure 6D,E).

Taken together, our results indicated that the M/TCs and GCs share similar input sources with different distribution patterns. Figure 7 summarizes the major input regions of the M/TCs and GCs in schematic form.

## 3. Discussion

The centrifugal projections of the OB provide a way in which ongoing information in higher brain areas can modulate early olfactory information processing [1,2,33]. Using virus-mediated tracing, we revealed the anatomical organization of centrifugal input to the OB, particularly for excitatory projection neurons, the M/TCs. Although the major brain regions we identified as presynaptic to the OB are consistent with previous studies [19,21,22], our experiment provided a whole-brain quantitative analysis of the input patterns and revealed several levels of specificity. We found that the input of the M/TCs was similar to that of the GCs, consistent with previous studies showing that different types of neurons in a given brain region receive afferent input from similar brain regions [34,35]. However, the centrifugal projections from the primary olfactory cortical areas and neuromodulatory areas to the OB are organized in different ways.

Moreover, consistent with previous tracing reports [17,20,29], we have validated that M/TCs, as well as GCs, receive notable inputs from the primary olfactory cortical areas including the AON and the PC, which are critical for odor identification and olfaction-related learning and memory [30,36,37,38]. Since the AON and PC send excitatory axons to the OB [10,19,39], activation of the projections of the AON and PC can directly excite GABAergic GCs in the OB, which, in turn, inhibit M/TCs from firing [1,17,21] and sometimes elicit a temporally precise increase in the probability of firing [21]. We also found that M/TCs integrated centrifugal input more specifically from the posterior AON and a small patch of the adjacent APC, whereas the GCs received broader input. Previous tracing results indicated that the AON maintains the dorsal–ventral topography of the OB [19,40,41,42], perhaps preserving the dorsal–ventral segregation of olfactory information. The dorsal AON is dedicated to innate responses, and the posteroventral AON is dedicated to learned responses [38,43]. The excitatory direct input to the M/TCs from the posterior AON, as well as the BF, raises the possibility that the ongoing olfactory processing in central brain areas may prime the M/TCs to fire when a predicted or desirable odor is detected in the learned behavior [44]. In contrast, the broad input to the GCs implies that the odor-evoked response in the OB is affected by diverse odor attributes and olfactory associations [19,40,41]. For instance, the centrifugal input to the GCs may underly odor localization [19,40,41], because GCs are heavily innervated by the contralateral AONpE, a region that precisely links symmetric olfactory maps of the bilateral OBs and thus is crucial in inter-bulbar communication [12,29,45].

Cholinergic input from the BF to the OB provides a way for attention and arousal to modulate olfactory processing by potentiating the adaptive behavior to environmental stimuli and decreasing the irrelevance behavior [2,33,46]. Despite a great deal of research [32,47,48,49], the role of cholinergic modulation on OB has not been well elucidated, partly because of the complex anatomy of cholinergic circuits [46,50,51]. Using trans-synaptic tracing tools, we revealed the cholinergic projections from the BF to the M/TCs, which verified that cholinergic projections exert their actions through synaptic transmission [52] in addition to extrasynaptic transmission [53]. Furthermore, we verified that individual cholinergic neurons of the BF can innervate both the M/TCs and GCs, extending previous anatomic studies [47,54]. Notably, there is a possibility that some cholinergic neurons of the BF that form synapses only with the M/TCs and not the GCs could be double-labeled with RV-DsRed and CTB647 in our dual-retrograde tracing experiment because CTB can be taken up by neurons via severed axons around the needle track, although this probability is small. This architecture resembles a broadcast model, in which individual cholinergic neurons broadcast information widely to coordinate the neuronal response [51,55], much more than the discrete output model [47], fitting well with the primary role of the cholinergic system in regulating the state of whole-brain neuronal networks. The diverse effects of cholinergic modulation on the OB observed in different studies [47,48,49] may be due to the diversity of target neuron populations and acetylcholine receptor subtypes, the layer-specific expression of the receptors, and the dynamic release of acetylcholine [23,51,55]. Future studies using optogenetic manipulation combined with electrophysiological recording in a cell-type-, projection-, or activity-specific manner, might reveal the precise role of cholinergic input in olfactory encoding and behavior.

Notably, when we compared the input patterns of two types of OB neurons in this study, different tracing methods were used, namely RV-EnvA-ΔG-DsRed in combination with helper AAV mediated retrograde trans-monosynaptic tracing, and RV-ΔG- mCherry retrograde tracing. The differences in the cellular tropism and transduction efficiency between the different viral tracing systems, which depend on the receptors on the targeted cells, the promoter of virus, etc., may influence the tracing efficiency to some degree [56,57,58]. Moreover, the efficiency of trans-synaptic spread, which depends on the G packaging into rabies virus particles, the uptake of rabies particles by presynaptic neurons, the levels of G expression in starter neurons, etc., may also influence the efficiency of tracing [59]. In addition, no tracing system can label all the presynaptic neurons with the same probability. Thus, in order to reduce the influence of the tracing systems, we compared the weighted input for the two tracing groups in the present study. Further research using viral tools with higher tracing efficiency combined with defined transgenic mouse lines will be needed to map the centrifugal input to different neuronal subtypes in the OB, thus helping us comprehensively understand the organizational principles of centrifugal projections to the OB.

In conclusion, our study elucidated the organizational principles of centrifugal projections to two different types of OB neurons, reflecting the complementary and coordinated strategies of information processing in the OB. Further research is required to better understand what kind of information the centrifugal projections transmit back to the OB, and how attention, emotion, and experience affect olfactory processing and behavior.

## 4. Materials and Methods

### 4.1. Animals and Stereotaxic Surgeries

Adult male C57BL/6J (Hunan SJA Laboratory Animal Co., Ltd, Changsha, China), Thy1-Cre (Jackson, Strain #006143, Barr Harbor, ME, USA) and ChAT-ires-Cre mice (Jackson, Strain #018957) were used. All surgical and experimental procedures were performed in accordance with the guidelines of the Animal Care and Use Committees in the Innovation Academy of Precision Measurement Science and Technology, Chinese Academy of Sciences, and all efforts were made to minimize the number and suffering of experimental animals.

The procedure for stereotaxic surgeries was similar to what we have used previously in Biosafety Level 2 animal facilities [34]. Briefly, the mice were anesthetized with 80 mg/kg sodium pentobarbital via intraperitoneal injection, then mounted in a stereotaxic holder (RWD, 68030, Shenzhen, China). The skull above one OB was removed, then the virus was injected into the targeted brain region with a glass micropipette connected to the syringe pump (Stoelting, 53311, Wood Dale, IL, USA).

### 4.2. Retrograde Tracing Strategies

For retrograde tracing from the M/TCs, 80 nL Cre recombinase-dependent helper viruses (a mixture of rAAV2/9-EF1α-DIO-GFP-TVA and rAAV2/9-EF1α-DIO-RVG, 1:1) were injected into the MCL/EPL of the medial (AP, +4.65 mm; ML, +0.30 mm; DV, −1.50 mm) or dorsolateral OB (AP, +4.65 mm; ML, +0.8 mm; DV, −1.10 mm) in Thy1-Cre mice. After 3 weeks, 120 nL RV-EnvA-ΔG-DsRed was injected into the same location of the OB. For retrograde tracing from the GCs, 80 nL RV-N2C(G)-ΔG-mCherry was injected into the GCL of the medial OB (AP, +4.65 mm; ML, +0.60 mm; DV, −1.50 mm) in C57BL/6J mice. In the dual-retrograde tracing experiment, retrograde monosynaptic viruses and retrograde tracer CTB647 (1%) were injected into the MCL and the GCL, respectively, of one OB in Thy1-Cre mice to label the input neurons of the M/TCs and GCs. After 6 days of RV injection, the animals were sacrificed for histology. The viruses used are listed in Table 1.

### 4.3. Histology and Imaging

Mice were overdosed with 100 mg/kg pentobarbital sodium via intraperitoneal injection and then perfused transcardially with 0.1 M phosphate-buffered saline (PBS), followed by 4% paraformaldehyde (PFA) in PBS. After post-fixation overnight in 4% PFA, the brain issues were placed in 30% sucrose in PBS for 48–72 h and then cut into 40 µm coronal slices using a cryostat microtome (Thermo Fisher, NX50, Waltham, MA, USA).

For neuronal identification, the brain slices were blocked for 1.5 hours with 10% normal donkey serum in PBS-T (0.3% Triton-X100), then incubated overnight with the primary antibodies (Anti-CaMKII alpha goat polyclonal antibody, Anti-ChAT goat polyclonal antibody, or Anti-GAD65/67 rabbit monoclonal antibody; 1:200–400) in 5% normal donkey serum, followed by species-specific fluorophore conjugated secondary antibodies (donkey anti-goat Alexa Fluor 488, donkey anti-goat Alexa Fluor 647, or donkey anti-rabbit fluorescein FITC; 1:250) for 1.5 h, and stained with DAPI (1.25 μg/mL) for 10 min. For the analysis of the distribution patterns, every sixth brain slice was selected and stained with DAPI. After mounting and sealing, the brain slices were imaged with a virtual microscopy slide scanning system (Olympus, VS 120, Tokyo, Japan) or a fluorescence laser scanning confocal microscope (Leica, TCS SP8, Buffalo Grove, IL, USA). The reagents used are in Table 2.

### 4.4. Whole-Brain Imaging

For labeling the cholinergic neurons of the BF, 80 nL of rAAV2/9-CAG-FLEX-GFP was injected into the BF (AP, +0.98 mm; ML, +0.00 mm; DV, −4.8 mm) of ChAT-ires-Cre mice. After 4 weeks of AAV injection, the animals were sacrificed for fluorescence micro-optical sectioning tomography (fMOST), using a similar procedure to that previously described [61]. Briefly, each mouse brain sample was immobilized in the AP direction on a 3D translation stage, and the whole-brain imaging system automatically performed the sectioning and imaging process to collect a brain-wide dataset.

### 4.5. Tracing Data Analysis

The division of the brain regions was mainly based on the Allen Brain Atlas. In order to quantify the spatial distribution of the input neurons, the number of virus-labeled input neurons in each brain area, subarea, or cell layer was counted for each using the cell counter plugin of ImageJ, then the weighted inputs were calculated. Furthermore, the position of each input neuron in a certain brain area (AON/PC/BF) was obtained using the ROI manager to calculate the weighted inputs along the AP, ML, and DV axes [62].

### 4.6. Reconstruction and Analysis of the Neurons

In line with a previous procedure [61], the raw image dataset was preprocessed and aligned with the Allen Reference Atlas. Single fluorescently labeled neurons were reconstructed interactively in Amira visualization and data analysis software (Visage Software, San Diego, CA, USA). For the analysis of the projection patterns, all the axon terminals of individual fluorescently labeled cholinergic neurons were marked, and their locations in different olfactory areas were calculated.

### 4.7. Statistical Analysis

Data are presented as the mean ± s.e.m., and significant differences are evaluated by two-tailed *t*-tests or one-way ANOVA followed by Tukey’s post hoc multiple comparison. The significance threshold was set at *p* < 0.05.

## Figures and Tables

**Figure 1 ijms-24-04579-f001:**
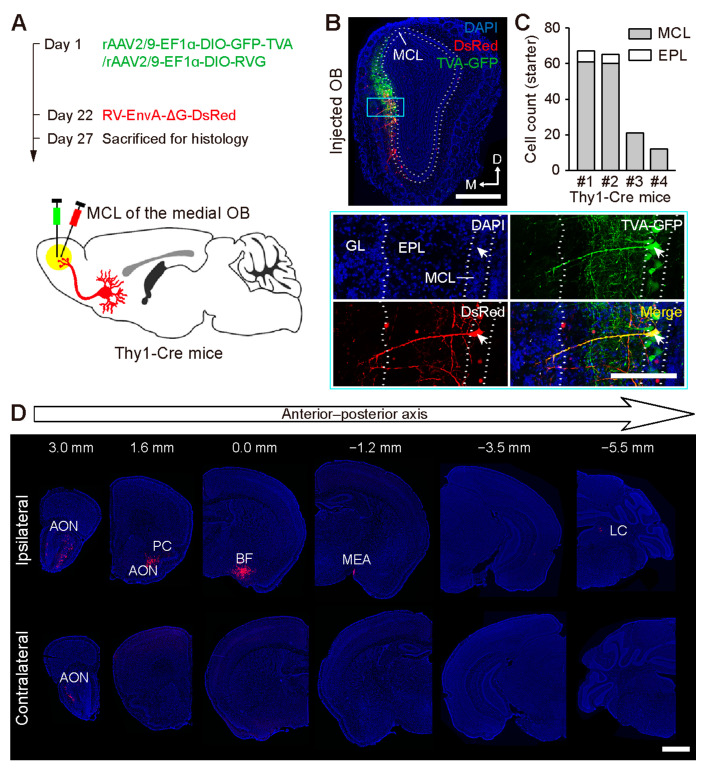
Retrograde tracing of the monosynaptic input to the M/TCs of the medial OB. (**A**) Strategy for tracing the input from the M/TCs using RV-based retrograde monosynaptic tracing viruses. (**B**) Upper: Representative image showing starter neurons in the MCL of the medial OB. Lower: Magnification of the blue outlined area in the upper image. The arrowhead indicates a starter neuron in the MCL co-expressing TVA-GFP and RV-DsRed. MCL, mitral cell layer; EPL, external plexiform layer; GL, glomerular layer. (**C**) Cell count of starter neurons in each mouse. (**D**) Examples of images showing fluorescently labeled input neurons in the major brain areas. AON, anterior olfactory nucleus; PC, piriform cortex; BF, basal forebrain; MEA, medial amygdalar nucleus; LC, locus coeruleus. Scale bars, 500 µm in (**B**) (upper); 200 µm in (**B**) (lower); 1 mm in (**D**).

**Figure 2 ijms-24-04579-f002:**
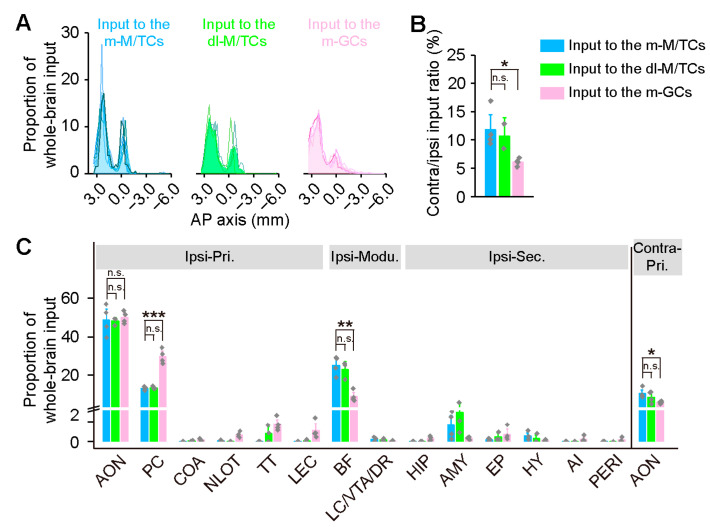
Brain-wide input distribution of the OB. (**A**) Quantified distribution of the input neurons targeting the M/TCs and GCs along the anterior–posterior (AP) axis. Colored lines denote individual animals. (**B**) Relative ratio of the contralateral input to the ipsilateral input. (**C**) Proportion of input neurons in discrete brain regions. Regions that contributed more than 0.1% of the total input are summarized here. Pri., primary olfactory areas; modu., neuromodulatory areas; sec., secondary olfactory areas; AON, anterior olfactory nucleus; PC, piriform cortex; COA, cortical amygdalar nucleus; NLOT, lateral olfactory tract nucleus; TT, taenia tecta; LEC, lateral entorhinal cortex; BF, basal forebrain; LC, locus coeruleus; VTA, ventral tegmental area; DR, dorsal raphe nucleus; HIP, hippocampal region; AMY, amygdalar area; EP, endopiriform nucleus; HY, hypothalamus; AI, agranular insular cortex; PERI, perirhinal cortex. Data are shown as the mean ± s.e.m.; gray rhombuses denote individual animals in (**B**,**C**). Dorsolateral M/TC: n = 3 animals; medial M/TC and medial GC: n = 4 animals per group. Unpaired two-tailed *t*-tests were used for (**B**,**C**). * *p* < 0.05, ** *p* < 0.01, *** *p* < 0.001, n.s. no significant difference.

**Figure 3 ijms-24-04579-f003:**
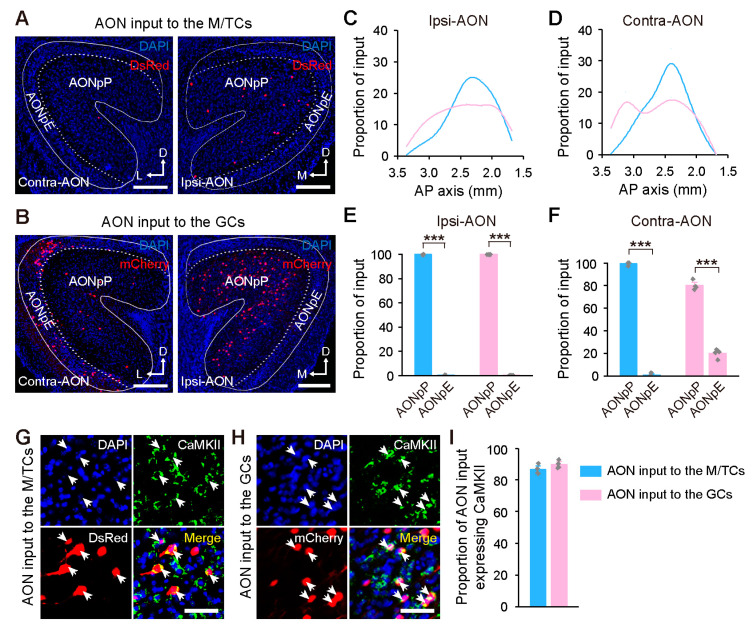
Organization of the centrifugal input from the AON to the OB. (**A**,**B**) Input neurons in the ipsilateral (left) and contralateral AONs (right) following retrograde tracing from the M/TCs (**A**) and GCs (**B**). (**C**,**D**) Cumulative distribution of the input neurons in the ipsilateral (**C**) and contralateral AONs (**D**) along the AP axis. (**E**,**F**) Subarea distribution of the input neurons in the ipsilateral (**E**) and contralateral AONs (**F**). Paired two-tailed *t*-tests, *** *p* < 0.001. (**G**,**H**) Identification of CaMKII-expressing input neurons in the ipsilateral AON (indicated by arrowheads) following retrograde tracing from the M/TCs (**G**) and GCs (**H**). (**I**) Proportion of CaMKII-expressing input neurons in the bilateral AONs. Data are shown as the mean ± s.e.m.; gray rhombuses denote individual animals in (**E**,**F**,**I**). Scale bars, 200 µm in (**A**,**B**); 50 µm in (**G**,**H**).

**Figure 4 ijms-24-04579-f004:**
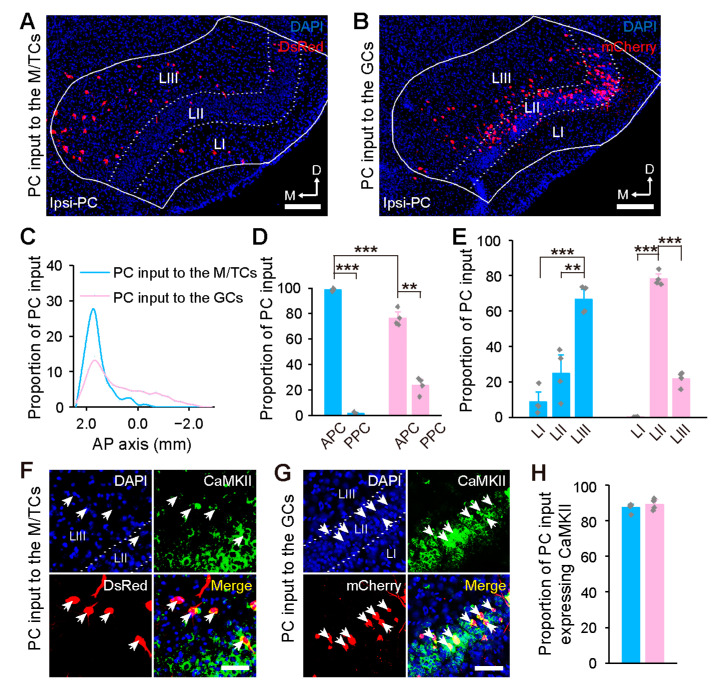
Organization of the centrifugal input from the PC to the OB. (**A**,**B**) Input neurons in the ipsilateral PC following retrograde tracing from the M/TCs (**A**) and GCs (**B**). (**C**) Cumulative distribution of the input neurons in the PC along the AP axis. (**D**) Subarea distribution of the input neurons in the PC. (**E**) Laminar distribution of the input neurons in the PC. (**F**,**G**) Identification of CaMKII-expressing input neurons in the PC (indicated by arrowheads) following retrograde tracing from the M/TCs (**F**) and GCs (**G**). (**H**) Proportion of CaMKII-expressing input neurons in the PC. Data are shown as the mean ± s.e.m.; gray rhombuses denote individual animals in (**D**,**E**,**H**). Scale bars, 200 µm in (**A**,**B**); 50 µm in (**F**,**G**). Two-tailed *t*-tests were used for (**D**); one-way ANOVA with post-hoc Tukey’s test was used for (E); ** *p* < 0.01, *** *p* < 0.001.

**Figure 5 ijms-24-04579-f005:**
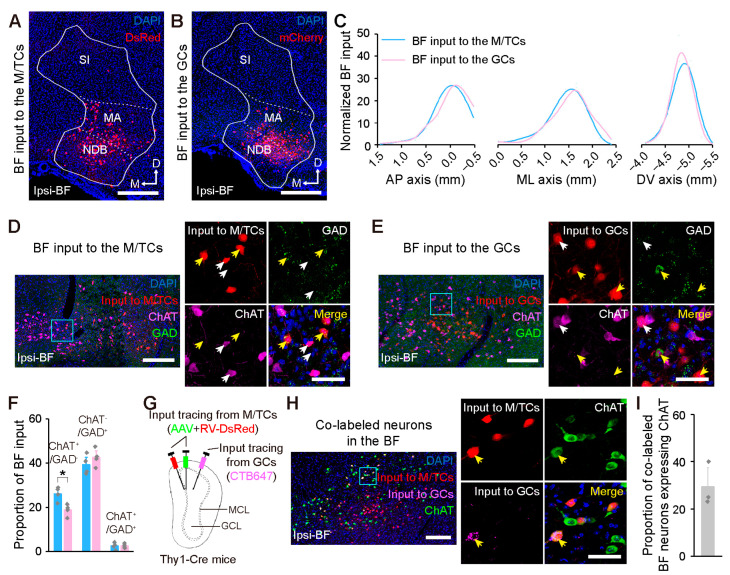
Organization of the centrifugal input from the BF to the OB. (**A**,**B**) Input neurons in the ipsilateral BF following retrograde tracing from the M/TCs (**A**) and GCs (**B**). NDB, diagonal band nucleus; MA, magnocellular preoptic nucleus; SI, substantia innominata. (**C**) Cumulative distribution of the input neurons in the BF along the AP, ML, and DV axes. (**D**,**E**) Left: Representative image showing ChAT-expressing and/or GAD65/67-expressing input neurons in the BF following retrograde tracing from the M/TCs (**D**) and GCs (**E**). Right: Magnification of the blue outlined area in the left-hand image. The white and yellow arrowheads indicate ChAT-expressing and GAD65/67-expressing input neurons, respectively. ChAT, choline acetyltransferase; GAD, glutamate decarboxylase. (**F**) Proportion of the BF input neurons expressing ChAT^+^/GAD^−^, ChAT^−^/GAD^+^, and ChAT^+^/GAD^+^. Unpaired two-tailed *t*-tests. * *p* < 0.05. n = 4 animals. (**G**) Strategy for dual-retrograde tracing from the M/TCs and GCs using two different colored retrograde tracers, namely the RV-based retrograde monosynaptic tracing viruses and the CTB647. (**H**) Left: Representative image of co-labeled neurons expressing ChAT in the BF. Right: Magnification of the blue outlined area in the left-hand image. The arrowhead indicates a ChAT-expressing co-labeled neuron. (**I**) Proportion of ChAT-expressing co-labeled neurons in the BF. n = 3 animals. Data are shown as the mean ± s.e.m.; gray rhombuses denote individual animals in (**F**,**I**). Scale bars, 500 µm in (**A**,**B**); 200 µm in (**D**) (left), (**E**) (left), and (**H**) (left); and 50 µm in (**D**) (right), (**E**) (right), and (**H**) (right).

**Figure 6 ijms-24-04579-f006:**
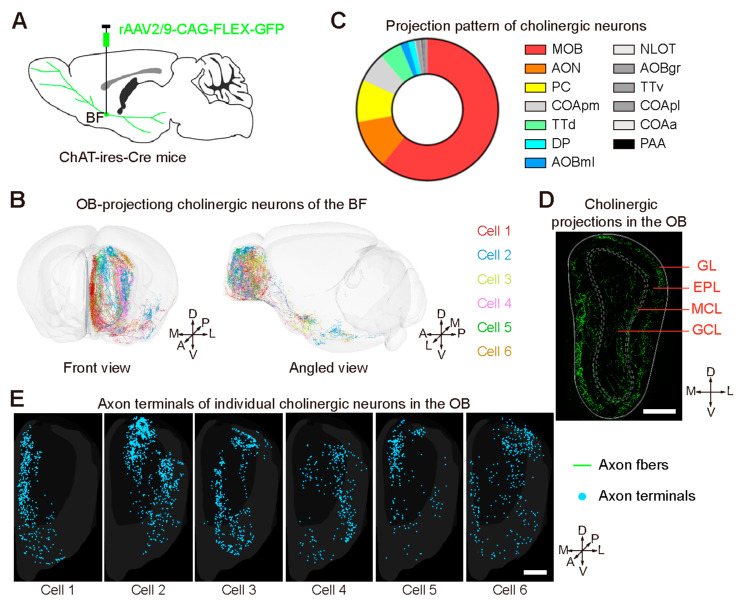
Axonal projections of the cholinergic neurons of the BF. (**A**) Strategy for labeling the cholinergic neurons of the BF using rAAV2/9-CAG-FLEX-GFP. (**B**) Six OB-projecting cholinergic neurons were reconstructed from the whole-brain database and are shown in different colors. (**C**) Distribution of the axon terminals of the cholinergic neurons in the primary olfactory areas. MOB, main olfactory bulb; COApm, posteromedial part of the cortical amygdalar nucleus; TTd, dorsal part of the taenia tecta; DP, dorsal peduncular cortex; AOBml, mitral cell layer of the accessory olfactory bulb; NLOT, nucleus of the lateral olfactory tract; AOBgr, granular cell layer of the accessory olfactory bulb; TTv, ventral part of the taenia tecta; COApl, posterolateral part of the cortical amygdalar nucleus; COAa, anterior part of the cortical amygdalar nucleus; PAA, amygdalopiriform transition area. (**D**) Representative image of fluorescently labeled cholinergic axon fibers in the OB. GL, glomerular layer; EPL, external plexiform layer; MCL, mitral cell layer; GCL, granular cell layer. (**E**) Axon terminals of individual cholinergic neurons in the OB. Scale bars, 500 µm in (**D**,**E**).

**Figure 7 ijms-24-04579-f007:**
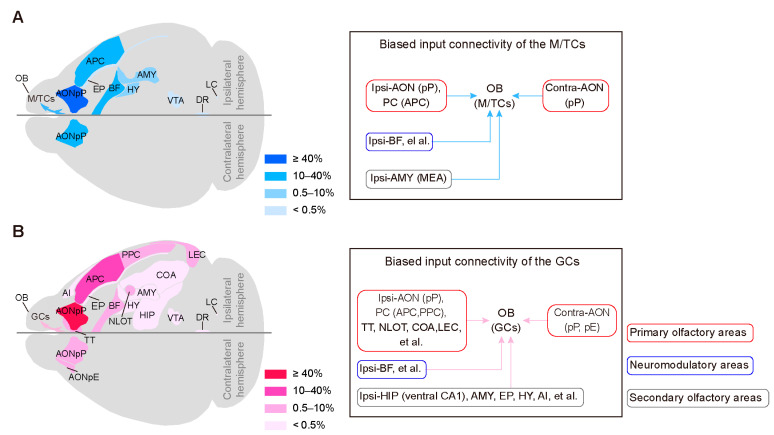
Summary of the major centrifugal input regions of the OB. (**A,B**) Schematic representation (left) and subnetwork organization (right) of the centrifugal input targeting M/TCs (**A**) and GCs (**B**). The color transparency represents the percentage of input to the MTCs (blue) and GCs (red). AON, anterior olfactory nucleus; AONpP, anterior olfactory nucleus pars principalis; AONpE, anterior olfactory nucleus pars externa; PC, piriform cortex; APC, anterior piriform cortex; PPC, posterior piriform cortex; TT, taenia tecta, NLOT, nucleus of the lateral olfactory tract; COA, cortical amygdalar nucleus; LEC, lateral entorhinal cortex, BF, basal forebrain; HIP, hippocampal region; AMY, amygdalar area, MEA, medial amygdalar nucleus; EP, endopiriform nucleus; HY, hypothalamus; AI, agranular insular cortex.

**Table 1 ijms-24-04579-t001:** Viruses used for retrograde tracing.

Virus Name	Resource	Titers
rAAV2/9-EF1α-DIO-GFP-TVA	Fuqiang Xu group [34,60]	6.50 × 10^12^ vg/mL
rAAV2/9-EF1α-DIO-RVG	Fuqiang Xu group [34,60]	5.00 × 10^12^ vg/mL
RV-EnvA-ΔG-DsRed	Fuqiang Xu group [34,60]	4.70 × 10^8^ PFU/mL
RV-ΔG-mCherry	Fuqiang Xu group [8,58]	6.00 × 10^7^ PFU/ML

**Table 2 ijms-24-04579-t002:** Reagents used for histology and imaging.

Antibodies, Recombinant Proteins, and Chemicals	Resource	Identification
Anti-CaMKIIα goat polyclonal antibody	Abcam	Cat# ab87597
Anti-ChAT goat polyclonal antibody	Millipore	Cat# ab144p
Anti-GAD65/67 rabbit monoclonal antibody	Abcam	Cat# ab183999
Donkey anti-goat Alexa Fluor 488	Jackson ImmunoResearch	Cat# 705-545-147
Donkey anti-goat Alexa Fluor 647	Jackson ImmunoResearch	Cat# 705-605-147
Donkey anti-rabbit fluorescein FITC	Jackson ImmunoResearch	Cat# 711-095-152
4’,6-Diamidino-2-phenylindole dihydrochloride (DAPI)	Beyotime	Cat# C1002
Paraformaldehyde (PFA)	Sigma	Cat# 158127
Cholera toxin subunit B (recombinant), Alexa Fluor 647 conjugate (CTB647)	Invitrogen	Cat# C34778

## Data Availability

The authors confirm that the data supporting the findings of this study are available within the article and its Appendix A.

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
