# Peer review of "Organizational Principles of the Centrifugal Projections to the Olfactory Bulb"

_ijms, 2023, doi:10.3390/ijms24054579_

Round 1
Reviewer 1 Report
The authors studied centrifugal projections in the olfactory system and revealed the organizational principles of centrifugal projections onto two different types of OB neurons, reflecting complementary and coordinated strategies of information processing in the OB. But the novelty is limited.
1. There are English grammar errors in the text which need to be further corrected.
2. The author stated that OB receives a substantial number of centrifugal inputs from central brain regions. We usually know is that smell transmits signals to the central brain regions through OB. So the author should have a good explanation.
3. How to identify the anatomical structure in the immunohistochemical map.
Author Response
Response 1: Thanks for your scrupulous review and valuable suggestions. We have checked our manuscript carefully and corrected the English grammar errors in the revised manuscript.
Response 2: Thank you for your helpful and constructive comments and suggestions. Previous works demonstrated that sensory input is a strong driver of neuronal activity in early sensory areas, but sensory information processing and transmission are not a simple feedforward process. Centrifugal input from central brain regions provides information about contextual factors, such as behavioral state, attention, or prior knowledge, that strongly modulate early sensory activity (Neuron. 2007; 54(5): 677-696). We have added in the Introduction in lines 47-51 that “OB receives primary afferent input from the sensory neurons, then processes and sends information to the primary olfactory cortex. At the meanwhile, OB receives a surprisingly large number of centrifugal inputs from the central brain regions, modulating olfactory processing and behavior”.
Response 3: Thank you for your helpful and constructive comments and suggestions. The high-magnification images of the immunohistochemical map in the manuscript were chosen after brain structure segmentation based on the Allen Brain Atlas. We have added the images with low magnification showing the anatomical structure in the immunohistochemical map, and marked the areas where high magnification images were chosen (Figure 6D, 6E, and 6H).
Reviewer 2 Report
the research is well disegned and the work is written clearly. The results are very interesting and full of data of interest.
Author Response
Response: Thank you for your strong support of our manuscript.
Reviewer 3 Report
I think that the experiments, taken together, help to understand the difficulty of the topic addressed. The bibliographic references are recently published. As the authors rightly conclude, further research will be needed to better understand what kind of information the centrifugal projections transmit back to OB.
Author Response
Response: We are grateful for your positive comments on our manuscript.
Reviewer 4 Report
This study investigated the centrifugal projections to neurons in the olfactory bulb (OB), mitral/tufted cells (M/TCs) and granule cells (GCs), using the virus-mediated retrograde monosynaptic tracing method. However, the results regarding centrifugal inputs to GCs largely overlapped with the results reported in previous studies, in particular in a study published by the same group (Wen et al. Bull. August, 2019, 35(4):709-723). Therefore, the novelty of this study should focus on the centrifugal inputs to M/TCs, which is shown more clearly than in previous work by using Thy1-Cre transgenic mice. The authors reported that mitral/tufted cells (M/TCs) receive strong inputs from the anterior olfactory nucleus (AON), piriform cortex (PC), and basal forebrain (BF). They also suggest that individual BF cholinergic neurons innervate multiple layers of the OB, forming synapses on both M/TCs and GCs. Overall, most of the data are clearly presented and the distribution in the brain of cells that synapses to M/TCs is worth reporting. The following are my comments that may need to be addressed prior to publication.
Major Comments:
1. Based on the results of labeling M/TCs in the medial (Figure 1) and dorsolateral (S1) OB, it is stated that "M/TCs in different parts of the OB received mostly similar inputs from central brain regions (lines 80-81)" and also "the M/TCs, even in different parts of the OB, integrate centrifugal inputs more specifically from the posterior AON and a small patch of adjacent APC (lines 244-246)". However, there are no quantitative studies of the results obtained from the labeling of dorsolateral M/TCs. Furthermore, it is possible that dorsal and ventral M/TCs receive centrifugal inputs in different ways. Therefore, the statement that M/TCs in different parts of the OB receive similar centrifugal inputs seems an overstatement and should be tempered.
2. The conclusion that individual BF cholinergic neurons synapse onto both M/TCs and GCs is based on the existence of ChAT+ neurons doubly labeled with DsRed (M/TCs) and CTB647 (GCs) (Figure 5H). However, CTB can be taken up by neurons via severed axons, and the injection procedure can cause axonal damage in the granule cell layer. Therefore, is it possible that ChAT+ neurons doubly labeled with DsRed and CTB647 in the BF synapse only to M/TCs and not to GCs? Further data or discussion supporting the author's conclusion would be needed.
The quality of Figure 6E is too low to show that individual BF neurons make synapses both in the mitral cell layer and the granule cell layer of the OB. Please provide better images. This may occur when converting the original images to PDF.
3. Since the methods for tracing M/TCs and GCs are different, I wonder how this difference affects the results. The authors also mention that "the difference of tracing efficiency and infection tropism between these two tracing systems may more or less influence the results (lines 279-280)". I suggest the authors to further discuss the possible influences.
Author Response
Response 1: Thanks for your scrupulous review and constructive suggestions. We have performed the quantitative analysis of the medial and dorsolateral M/TCs labeling data. The results showed that there was no significant difference between the medial and dorsolateral OB tracing groups. We have added charts in Figure 2 and modified the statement in the revised manuscript in lines 103-106 that “M/TCs in the medial and dorsolateral parts of the OB received generally similar input from the central brain regions. Whether dorsal and ventral M/TCs receive centrifugal input in similar ways needs to be further investigated.” and in lines 251-253 that “we also found that the M/TCs integrate centrifugal input more specifically from the posterior AON and a small patch of the adjacent APC, whereas the GCs received broader input”.
Response 2: Thanks for your insightful suggestion. We agree with you on CTB can be taken up by neurons via severed axons around the needle track, and perhaps some BF cholinergic neurons doubly labeled with DsRed and CTB647 synapse only to M/TCs and not to GCs. However, we think this is a small probability event for the following reasons: (1) The diffusion range of CTB in our tracing study (about hundreds of microns) is much larger than the diameter of the needle passage (about ten microns), and there is no obvious CTB leakage signal around the MCL of injected OB, thus the proportion of input neurons marked by CTB leakage should be very low (about one of ten thousand). In fact, this proportion is far lower than the proportion of BF cholinergic neurons doubly labeled with DsRed and CTB647 to CTB-labeled BF neurons (3.33% ± 0.87%; n = 3 animals). The majority of CTB-labeled input neurons should target GCs. (2) Moreover, we verified that the OB-projecting BF cholinergic neurons sent their axon terminals to multiple layers of the OB including the MCL, EPL and GCL (lines 202-204). Combing with previous reports (Nat Commun. 2017; 8(1): 652; J Neurosci 1986; 6(1): 281-292), we suppose that individual BF cholinergic neurons innervate both M/TCs and GCs (lines 273-274). Furthermore, figure 6E has been changed to high-resolution images in the revised version.
Response 3: We agree with this helpful suggestion. We have added in the revised Discussion in lines 286-298 that “The differences in cellular tropism and transduction efficiency among different viral tracing systems, which depend on the receptors on the targeted cells, the promoter of the virus, etc., may influence the tracing efficiency more or less (J Neurosci 2015, 35(24): 8979-85; Neurosci Bull 2020, 36(3): 202-216). Moreover, the efficiency of transsynaptic spread, which depends on the G packaging into rabies virus particles, the uptake of rabies particles by presynaptic neurons, the levels of G expression in starter neurons, etc., may also influence the tracing efficiency (Neuron 2016, 89: 711–724). In addition, no tracing system may label all presynaptic neurons with the same probability. Thus, in order to reduce the influence by the tracing systems, we compared the weighted input of each brain region (the proportion of the number of labeled neurons within a certain brain region in the total number of the input neurons in the whole brain except the injected OB) for the two tracing groups in the present study. Further researches using viral tools with higher tracing efficiency combined with defined transgenic mouse lines will be needed to map the centrifugal inputs to different neuronal subtypes in the OB, thus help us comprehensively understand the organizational principles of centrifugal projections to the OB.”
Round 2
Reviewer 4 Report
Thank you very much for revising your manuscript. My comments were addressed, and the manuscript has been much improved.
I would like to suggest adding to the manuscript a discussion of “there is a possibility that some BF cholinergic neurons that synapse only to M/TCs and not to GCs could be double-labeled with DsRed and CTB647 because CTB can be taken up by neurons via severed axons around the needle track, although this is a small probability”. Other than that, I have no further comments.
Author Response
Response 1: Thanks for your insightful suggestion. We have added in the revised Discussion in lines 272-275 that “Notably, there is a possibility that some BF cholinergic neurons that synapse only to M/TCs and not to GCs could be double-labeled with RV-DsRed and CTB647 in our du-al-retrograde-tracing experiment because CTB can be taken up by neurons via severed axons around the needle track, although this is a small probability.”